# Interactions Between Spermine-Derivatized Tentacle Porphyrins and The Human Telomeric DNA G-Quadruplex

**DOI:** 10.3390/ijms19113686

**Published:** 2018-11-21

**Authors:** Navin C. Sabharwal, Jessica Chen, Joo Hyun (June) Lee, Chiara M. A. Gangemi, Alessandro D’Urso, Liliya A. Yatsunyk

**Affiliations:** 1Department of Chemistry and Biochemistry, Swarthmore College, Swarthmore, PA 19081, USA; navin.sabharwal.424@gmail.com (N.C.S.); ymchen.017@gmail.com (J.C.); jlee2143@gmail.com (J.H.(J.)L.); 2Lerner College of Medicine, Cleveland Clinic, Cleveland, OH 44195, USA; 3School of Dental Medicine, University of Pennsylvania, Philadelphia, PA 19104, USA; 4College of Dentistry, New York University, New York, NY 10010, USA; 5Department of Chemical Science, University of Catania, 95125 Catania, Italy; gangemichiara@unict.it

**Keywords:** G-quadruplex, tentacle porphyrins, Zn(II) porphyrin, anti-cancer therapy, end-stacking

## Abstract

G-rich DNA sequences have the potential to fold into non-canonical G-Quadruplex (GQ) structures implicated in aging and human diseases, notably cancers. Because stabilization of GQs at telomeres and oncogene promoters may prevent cancer, there is an interest in developing small molecules that selectively target GQs. Herein, we investigate the interactions of *meso*-tetrakis-(4-carboxysperminephenyl)porphyrin (TCPPSpm4) and its Zn(II) derivative (ZnTCPPSpm4) with human telomeric DNA (Tel22) via UV-Vis, circular dichroism (CD), and fluorescence spectroscopies, resonance light scattering (RLS), and fluorescence resonance energy transfer (FRET) assays. UV-Vis titrations reveal binding constants of 4.7 × 10^6^ and 1.4 × 10^7^ M^−1^ and binding stoichiometry of 2–4:1 and 10–12:1 for TCPPSpm4 and ZnTCPPSpm4, respectively. High stoichiometry is supported by the Job plot data, CD titrations, and RLS data. FRET melting indicates that TCPPSpm4 stabilizes Tel22 by 36 ± 2 °C at 7.5 eq., and that ZnTCPPSpm4 stabilizes Tel22 by 33 ± 2 °C at ~20 eq.; at least 8 eq. of ZnTCPPSpm4 are required to achieve significant stabilization of Tel22, in agreement with its high binding stoichiometry. FRET competition studies show that both porphyrins are mildly selective for human telomeric GQ vs duplex DNA. Spectroscopic studies, combined, point to end-stacking and porphyrin self-association as major binding modes. This work advances our understanding of ligand interactions with GQ DNA.

## 1. Introduction

DNA can exist in a variety of secondary structures [1] in addition to the right-handed double-stranded (dsDNA) form first proposed by Watson and Crick in 1953. One example is G-Quadruplex (GQ) DNA, a non-canonical DNA structure formed by guanine rich sequences [2]. The primary structural unit of GQ DNA is a G-tetrad which consists of four guanines associated through Hoogsteen hydrogen bonding (Figure 1A). G-tetrads interact with each other via π-π stacking, and are linked by the phosphate sugar backbone, forming GQs. The stability of the GQ is further enhanced by coordinating cations [3,4]. In fact, biological GQs with 2–4 G-tetrads would not fold without a cation due to a strong repulsion of guanine carbonyls in the center of each tetrad (Figure 1A). Unlike dsDNA, GQs exhibit high structural diversity, adopting parallel, mixed-hybrid, and antiparallel topologies (Figure 1B). Bioinformatics studies suggest that sequences with GQ-forming potential are prevalent in highly-conserved functional regions of the human genome including telomeres, oncogene promoters, immunoglobulin switch regions, and ribosomal DNA [5,6,7,8], and may regulate numerous biological processes. Evidence for GQ formation inside the cell was recently presented [9,10,11,12], and studies are underway to better assess their in vivo roles [2].

Telomeres protect the ends of eukaryotic chromosomes from degradation and fusion and contain tandem repeats of dTTAGGG [13]. The 22-mer human telomeric DNA sequence dAGGG(TTAGGG)_3_ (Tel22) is well-studied and has been shown to form diverse GQ structures in vitro [14,15,16], see Figure 1B. The topology, stability, and homogeneity of the human telomeric DNA depends on the DNA length and the identity of the nucleotides at 5′ and 3′ ends. In addition, the nature of the central stabilizing cation, the presence of small molecules, annealing temperature and rate, and molecular crowding reagents impact the resulting secondary structure. In K^+^, Tel22 forms a parallel GQ with three G-tetrads and three TTA propeller loops, but only in the presence of molecular crowding conditions [17,18], some small molecules (e.g., N-methylmesoporphyrin IX, NMM) [19,20], under crystallization conditions [21], or at high DNA concentration [22]. In Na^+^, Tel22 adopts an antiparallel topology with three G-tetrads connected by two lateral loops and one central diagonal loop [23]. In the dilute K^+^ solutions favored in this work, Tel22 adopts at least two (3 + 1) mixed-hybrid structures called Form 1 and Form 2 [24,25,26,27,28]. The two forms have one propeller loop and two lateral loops, but differ by loop orders; three G-rich strands run in the same direction and opposite from that of the fourth strand, hence the name (3 + 1). Other GQ topologies exist under these conditions (e.g., an antiparallel GQ with two G-tetrads) [29], but at low abundance. It has been proposed that formation of GQ structures at telomeres inhibits the activity of telomerase, the enzyme responsible for maintenance of telomeres integrity, leading to cell immortality. Because telomerase is upregulated in 85–90% of cancers [30], stabilization of GQs by small molecule ligands has emerged as a novel, selective, anti-cancer therapeutic strategy [31,32]. 

Porphyrins are one of the earliest classes of DNA ligands. Their interactions with GQ DNA were first studied in 1998 [33], and with dsDNA as far back as 1979 [34], and are still of great interest [35]. Porphyrins are aromatic, planar, and the size of their macrocycle (~10 Å) matches that of a G-tetrad (~11 Å), leading to an efficient π-π stacking. Cellular uptake and localization studies demonstrate that porphyrins accumulate rapidly in nuclei of normal and tumor cells [36,37] at levels sufficient for tumor growth arrest; yet they are non-toxic to somatic cells [38]. Porphyrins can be readily functionalized to optimize their GQ-stabilizing ability and selectivity, solubility, and cell permeability. Our laboratory and others have characterized binding of numerous porphyrins, including NMM [19,20,39], *meso*-tetrakis-(*N*-methyl-4-pyridyl) porphyrin (TMPyP4) [38,40], and its various derivatives [41,42,43,44] to human telomeric DNA. Porphyrins can bind to GQ DNA via end-stacking, which has been characterized spectroscopically [45,46], and observed in structural studies [20,47]. Intercalation has been suggested [46,48,49,50], but is considered energetically unfavorable for short GQs with 2-4 G-tetrads. Porphyrins can also interact with the grooves [51] and loops [52] of GQs. Porphyrin metallation is expected to enhance its GQ binding due to the electron-withdrawing property of the metal, which reduces the electron density on the porphyrin, improving its π-π stacking ability. The enhancement of porphyrin’s binding to GQ is especially strong when the metal is positioned above the ion channel of the GQ. 

In this work, we focus on two novel tentacle porphyrins, *meso*-tetrakis-(4-carboxyspermine-phenyl)porphyrin, TCPPSpm4 and its Zn(II)-derivative, ZnTCPPSpm4, Figure 1C. Binding of tentacle porphyrins to dsDNA is well studied [53,54,55,56], but their interactions with GQ DNA remain poorly characterized. We introduced spermine groups to enhance the GQ-binding potential, solubility, and biocompatibility of the porphyrins. Polyamines have been reported to interact with DNA by both electrostatic forces and via site-specific interactions with the phosphate backbone and DNA bases [57,58,59]. In some cases polyamines induced conformational modifications [60]. Spermine was shown to preferentially bind to the major groove of dsDNA [59]. A variety amines (e.g., pyrrolidine, piperidine, morpholine, 1-ethylpiperazine, *N*,*N*-diethylethylenediamine, and guanidine) have been incorporated into GQ ligands, leading to improvements in their GQ binding affinities and water solubility [61,62,63,64,65]. Of equally strong importance, spermine is essential for cellular growth, differentiation [66], and protection against double-strand breaks. Polyamines are currently being exploited as a transport system for cancer drugs due to their well-known ability to accumulate in neoplastic tissues [67,68,69,70,71]. Therefore, we added spermine to *meso*-tetrakis-(4-carboxyphenyl)porphyrin not only to improve its GQ-binding, but also to facilitate its delivery to cancer cells in future biological studies. 

We characterized the interactions between human telomeric DNA and TCPPSpm4 or ZnTCPPSpm4 in a K^+^ buffer through UV-Vis, fluorescence, and circular dichroism (CD) spectroscopies, resonance light scattering (RLS), and fluorescence resonance energy transfer (FRET) assays. We demonstrate that both porphyrins bind tightly to Tel22 GQ with a high binding stoichiometries (2–4:1 for TCPPSpm4 and 10–12:1 for ZnTCPPSpm4) and stabilize it strongly with mild selectivity over dsDNA. Our data are consistent with end-stacking binding mode and DNA-assisted porphyrin self-stacking. 

## 2. Results and Discussion

In this work, we focus on two tentacle porphyrins, *meso*-tetrakis(4-carboxysperminephenyl) porphyrin, TCPPSpm4, and its Zn(II) derivative, ZnTCPPSpm4. Both porphyrins are modified with four spermine arms, see Figure 1C. The *pKa* of the spermine amine groups in TCPPSpm4 was measured to be ~5.8 for the first protonation and ~8 for the second protonation [72]. Therefore, this porphyrin is expected to be at least tetracationic at pH 7.2 used in this work. Zn(II) was introduced into TCPPSpm4 to improve its GQ binding due to electron-poor nature of the metal. In addition, Zn(II) is coordinated to an axial water, which is expected to prevent its intercalation into dsDNA, and thus, to improve its selectivity. Binding of TCPPSpm4 to the GQ aptamer (dTGGGAG)_4_ was recently characterized [73], whereas binding of ZnTCPPSpm4 to any of the GQs has not yet been tested. Here, we explore in detail how both porphyrins interact with human telomeric GQ DNA, Tel22. 

### 2.1. UV-Vis Spectroscopy Demonstrates that TCPPSpm4 and ZnTCPPSpm4 Bind Tightly to Tel22

Due to the excellent chromophoric properties of both porphyrins, their binding to Tel22 was monitored using Soret band of 415 nm for TCPPSpm4 and 424 nm for ZnTCPPSpm4. We first performed a dilution study which indicated that the porphyrins maintain their aggregation state, assumed to be monomeric, in the concentration range of 1–40 µM (Appendix A). Subsequently, both porphyrins were titrated with Tel22; representative UV-Vis titrations are shown in Figure 2. The extinction coefficient for the TCPPSpm4-Tel22 complex was determined to be (1.2 ± 0.2) × 10^5^ M^−1^cm^−1^ at 429 nm and (0.54 ± 0.04) × 10^5^ M^−1^cm^−1^ for ZnTCPPSpm4-Tel22 at 435 nm. The Soret band of TCPPSpm4 displayed a pronounced red shift (Δλ) of 13.5 ± 0.5 nm and hypochromicity (%H) of 58 ± 6 % upon addition of Tel22. The corresponding values for ZnTCPPSpm4 are similar with Δλ of 11.3 ± 0.6 and % H of 58 ± 5%. Red shift of ~15 nm and %H of ~50% were obtained for TCPPSpm4 binding to another GQ structure formed by (dTGGGAG)_4_ aptamer [73]. High values of Δλ and %H indicate strong interactions between the π-systems of porphyrins and GQ, characteristic of either end-stacking or intercalation. Pasternack et al. found that intercalation of a porphyrin into dsDNA can be identified by %H > 40% and Δλ ≥ 15 nm [74]. Although supported by molecular dynamics stimulation studies [50], this mode of binding has not yet been detected in structural studies. On the other hand, both end-stacking [20,47] and loop binding [52] have been observed in X-ray structures of porphyrin-GQ complexes. 

To extract binding constants, we employed the Direct Fit method, which is the simplest way of treating the titration data, as it assumes equivalent and independent binding sites. Such data treatment is justified by the presence of the isosbestic points, yet it is an oversimplification in view of high stoichiometric ratios obtained (see below) and the presence of detectable shoulders, especially in final samples. Data analysis yielded a binding constant, *Ka*, of (4.7 ± 0.7) × 10^6^ M^−1^ for TCPPSpm4 assuming a binding stoichiometry of 4:1; and *Ka* of (1.4 ± 0.7) × 10^7^ M^−1^ for ZnTCPPSpm4 assuming a binding stoichiometry of 12:1. The high *Ka* values indicate strong binding between Tel22 and the porphyrins and correlate well with the high values of Δλ and %H. ZnTCPPSpm4 binds three times tighter than its free-base analogue, possibly due to the presence of electron withdrawing metal. This binding is likely further enhanced by electrostatic attractions due to high charges on the porphryins and by interactions of four spermine arms with the grooves of Tel22 GQ. 

To independently verify the stoichiometry for porphyrin-Tel22 binding, we used Job’s method, also known as the method of continuous variation [75]. In this method, the mole fraction of DNA and porphyrin is varied while their total concentration is kept constant. The mole fraction at the maximum or minimum on the plot of absorbance vs mole fraction corresponds to the binding stoichiometry between the two binding partners [76]. Representative Job plots are depicted in Figure 3. Job plot experiments for TCPPSpm4-Tel22 system yielded an average mole fraction of 0.70 ± 0.04, which corresponds to the binding of 2–3 porphyrins to one Tel22. For the ZnTCPPSpm4-Tel22 system, Job plot yielded a mole fraction value of ~0.9, which corresponds to the binding of nine porphyrin molecules to one Tel22 GQ. In both cases, binding stoichiometries are somewhat lower than those obtained via fitting of the UV-vis titration data. Similar discrepancy was also observed in our previous work where we investigated binding of four different cationic porphyrins to two parallel GQs [77]. Job plot stoichiometry is lower because it represents only the major binding event, while stoichiometry obtained via fitting of UV-vis titration data encompasses strong, weak, and non-specific binding. It is also important to remember that binding stoichiometries of 1:1 and 2:1 can be clearly differentiated via Job’s method, but higher binding stoichiometries are difficult to determine precisely. For example, binding ratios of 4:1 and 5:1 correspond to molar fractions of 0.8 and 0.83, respectively, which would likely be impossible to distinguish, given the expected level of data accuracy. The unusually high binding stoichiometry supports the involvement of multiple binding modes such as end-stacking, electrostatic interactions, and groove binding, the latter two resulting from the presence of spermine arms. It also suggests the possibility of porphyrin self-association on the DNA backbone. The much higher binding stoichiometry for ZnTCPPSpm4 is puzzling, especially in light of ZnTCPPSpm4′s axial water molecule, which is expected to inhibit some binding modes, such as porphyrin self-association. However, slipped self-stacking is still possible. 

### 2.2. RLS Indicates the Formation of Discrete Stoichiometric Porprhyrin-Tel22 Complexes

Because UV-vis titrations yielded high stoichiometry for porphryin-Tel22 complexes, we employed the RLS method [78] to check for possible aggregation. In RLS, porphyrin solution is excited close to its Soret maximum and the scattering is measured at the same wavelength. If aggregated (alone or on a substrate), porphyrins display enhanced Rayleigh scattering originating from electronic coupling between the individual molecules in the assembly. To detect communication between porphyrins, RLS experiments are performed under porphyrin excess, unlike UV-vis titrations, where DNA excess is used.

The RLS intensity of TCPPSpm4 alone is low (Figure 4A), indicating an absence of aggregation in agreement with UV-vis dilution studies (Appendix A). The addition of Tel22 does not change the RLS signal in the [TCPPSpm4]/[Tel22] range of 40–8. Below this ratio, however, RLS signal starts to increase and reaches a maximum at [TCPPSpm4]/[Tel22] = 2, suggesting the formation of an assembly with strong electronic communication between porphyrins. Further addition of Tel22 does not change RLS, indicating that the TCPPSpm4-Tel22 complex is stable. Adding more Tel22 to this solution eventually leads to drastic decline in RLS signal, owing to the precipitation of the complex (data not shown).

ZnTCPPSpm4 does not aggregate alone or in the [ZnTCPPSpm4]/[Tel22] range of 40–14 (Figure 4B). When more Tel22 is added, however, stable aggregates are formed at [ZnTCPPSpm4]/[Tel22] ~ 13, in line with the stoichiometry determined in UV-vis experiments. Subsequent addition of Tel22 does not change RLS until [ZnTCPPSpm4]/[Tel22] ~ 2, at which point the RLS signal rises up to 1:1 ratio then starts to decrease, although the observed changes are small.

Taken together, the RLS data allow us to (i) exclude porphyrin aggregation in the absence of DNA; (ii) confirm formation of discrete porphryin-Tel22 complexes with a stoichiometry consistent with that measured in UV-vis; and (iii) exclude existence of large, non-stoichiometric porphryin-Tel22 aggregates. Overall, RLS and UV-vis data support our hypothesis of DNA-assisted porphyrin self-aggregation on Tel22 which leads to strong electronic communication between individual porphyrins in the assembly.

### 2.3. Fluorescence of TCPPSpm4 and ZnTCPPSpm4 Decreases in the Presence of Tel22 Suggesting DNA-Assisted Porphyrin Self-Association

The steady-state fluorescence emission spectrum of a porphyrin is produced by the first excited state, S_1_, and the charge-transfer state between the porphyrin ring and its peripheral substituents (in this case carboxysperminephenyl groups). The coupling between these two states leads to quenching of the fluorescence signal, which occurs in polar solvents or when the rotation of peripheral substituents is unrestricted. TCPPSpm4 fluoresces in aqueous solution, producing a peak at 643 nm and a shoulder at 702 nm, as has been previously observed [72]. At the same time, ZnTCPPSpm4 produces a split peak at 607 and 657 nm, Figure 5, suggesting that the rotation of its side-chains is more restricted.

Position and intensity of the fluorescence peak of a porphyrin is strongly sensitive to its environment and, thus, can report on porphyrin binding to GQ DNA [79]. Addition of Tel22 GQ to TCPPSpm4 leads to a dramatic decrease in fluorescence intensity and a red shift of 10 and 15 nm for the 643 and 702 nm peaks, respectively. The spectra at saturating amount of Tel22 are sharper and better resolved, Figure 5A, suggesting restriction in rotation of the peripheral groups upon GQ binding. Similarly, the fluorescence intensity of ZnTCPPSPm4 decreased dramatically upon addition of Tel22, but the red shift observed was significantly smaller, i.e., 5 and 3 nm for the 607 and 657 nm bands, respectively. In both cases, the original dramatic decrease in signal intensity is followed by a small increase in the signal at high [Tel22]/[porphyrin] ratios (see Appendix A) suggesting a change in a mechanism of ligand interactions with Tel22 or with each other. The strong decrease in fluorescence could be explained by close interactions between porphyrins and Tel22 as well as by self-association of porphyrins assisted by the DNA backbone. Such interpretation is consistent with reported high binding stoichiometry, especially for ZnTCPPSpm4. Similar to our case, the steady-state fluorescence of the Zn(II) derivative of a widely-studied porphyrin, TMPyP4, decreased upon addition of tetrastranded parallel GQs [77] and poly(dG-dC) dsDNA [80], although in both cases the decrease was not as dramatic as in the present case. 

### 2.4. FRET Studies Indicate that Both Porphyrins Have Exceptional Stabilizing Ability and Modest Selectivity toward Tel22 GQ 

FRET is a benchmark technique in the quadruplex field enabling facile and reliable measurement of ligands’ stabilizing ability and selectivity for GQ DNA [81]. We used F21D, a 21-nt sequence of the human telomeric DNA labeled with 6-FAM fluorescent dye at the 5′ end and a quencher, Dabcyl, at the 3′ end (5′-6-FAM-GGG(TTAGGG)_3_-Dabcyl-3′). We have thoroughly characterized the fold and stability of this sequence in our earlier work [19]. The addition of up to 7.5 eq. of TCPPSpm4 and up to 20 eq. of ZnTCPPSpm4 to F21D resulted in a concentration-dependent increase in Tm of F21D by 36 ± 2 °C and 33 ± 2 °C, respectively (Figure 6A; raw data are shown in Appendix A). Our data shows that both porphyrins stabilize Tel22 GQ to a great extent, but the stabilization curve for ZnTCPPSpm4 is sigmoidal, and only weak stabilization is observed up to 1.6 µM (8 eq.) of the porphyrin. This data is in agreement with high stoichiometry of the ZnTCPPSpm4-Tel22 complex determined in UV-vis and Job plot studies.

Selectivity is an essential characteristic of an ideal anticancer GQ ligand, because a drug that binds readily to dsDNA will require a greater concentration to achieve its therapeutic effect, or even cause cytotoxicity. Thus, we conducted FRET competition studies in the presence of large excess of CT DNA and a fixed ligand concentration (Figure 6B). The selectivity ratio, defined as the fold of competitor necessary to reduce ΔTm by 50%, was calculated to be 270 for TCPPSpm4 and 200 for ZnTCPPSpm4. While the porphyrins prefer GQ to dsDNA, the observed selectivity ratios are rather modest. Such modest selectivity is likely due to strong electrostatic interactions between the positively charged porphyrins and negatively charged DNA (GQ, dsDNA, etc). This hypothesis is supported by our earlier work showing that reducing the charge on a porphyrin increases its selectivity for GQ DNA [44]. Our laboratory previously demonstrated that another Zn(II)-metallated porphyrin, ZnTMPyP4, displays selectivity ratio of 100 toward F21D vs CT DNA, while its free-base analogue displays a selectivity ratio of 300 [42]. These values are on the same scale and display the same trend as the one obtained in this work. Overall, FRET studies suggest that both porphyrins are robust stabilizers of human telomeric DNA, with TCPPSpm4 displaying both superior selectivity and stabilizing ability. 

### 2.5. Circular Dichroism (CD) Signal Decreases upon Addition of Porphyrins Signifying Interaction between Porphyrins and Tel22

To determine if porphyrin binding alters the topology of the Tel22 GQ, we performed CD annealing and titration studies. CD is an excellent method to report on the type of GQ fold and its alteration upon ligand binding. The CD signature of Tel22 in potassium buffer (5 mM KCl) is well characterized in our previous works [19] and that of others [22], and contains a peak at 295 nm and a shoulder at ~250 nm. Titration of TCPPSpm4 under kinetic conditions (with short 12 min equilibration) did not alter the conformation of Tel22, but lead to dramatic decrease in the intensity of 295 nm peak (Figure 7A). Under similar conditions, ZnTCPPSpm4 caused only a mild decrease of CD signal intensity (Figure 7B). To investigate the system under thermodynamic equilibrium, Tel22 samples were annealed with ~ 2 eq. of porphyrins and equilibrated overnight. The CD signals displayed stronger decrease (Figure 7C,D), in part caused by minor precipitation. Decrease in CD signal intensity was also observed upon interaction of TCPPSpm4 with (dTGGGAG)_4_ GQ aptamer [73]. Other metallated porphyrins, such as PtTMPyP4 [43], CuTMPyP4, and NiTMPyP4 [82] caused decrease in the intensity of Tel22 CD signal in potassium buffer, while CoTMPyP4 and ZnTMPyP4 did not [82]. 

The porphyrin-induced decrease in CD signal intensity could be explained, in part, by DNA precipitation, most likely caused by highly charged spermine arms of the porphyrin ligands. The precipitation was minor and was only observed at high porphyrin and DNA concentrations (above 10 µM DNA). In addition, the observed behavior in CD titrations could be explained by preferential binding of porphyrins to single-stranded (ssDNA), which disfavors GQ in the GQ DNA ↔ ssDNA equilibrium. This mode of binding was observed for TMPyP4 [83], triarylpyridines [84], and anthrathiophenedione [85]. However, such data interpretation seems to contradict the observed stabilization of human telomeric DNA in our FRET studies (Figure 6A). Alternatively, we can explain the observed decrease in CD signal intensity by proposing that porphyrins bind to GQ DNA by disrupting and replacing one or more of the G-tetrads, leading to unchanged or even enhanced stability. Such explanation reconciles our CD and FRET data and was first proposed by Marchand et al. on the basis of an extensive CD and mass spectrometry study [86].

### 2.6. The Presence of Induced CD (iCD) Confirms Close Contacts between Porphyrins and Tel22 Aromatic Systems

We further characterized porphyrin-Tel22 interactions by investigating changes in the CD Soret region. Chromophoric but achiral porphyrins produce no CD signal in this region, and the DNA CD signal is found exclusively in the UV region. However, when DNA and porphyrin interact, the complex is both chiral and chromophoric, and will produce an iCD when the π-system of a porphyrin is in close proximity to that of the DNA. For ligand binding to duplex DNA, the type of iCD has been found to correlate with the binding mode: a positive iCD corresponds to external binding and a negative one indicates intercalation [87,88]. However, a similar correlation has not yet been established for porphyrin-GQ interactions due to the scarcity of empirical data on binding modes other than end-stacking. 

The addition of Tel22 to each porphyrin at stoichiometric amounts yielded a bisignate iCD with a strong positive component (Figure 8). The trough and the peak occur at 410 and 426 nm for TCPPSpm4-Tel22 and at 427 and 442 nm for ZnTCPPSpm4-Tel22, which is consistent with their Soret band positions. Once we established the presence of the iCD, we conducted CD titrations in the Soret region. Due to low iCD signal intensity, the data display high variability, but nevertheless indicate that the strongest iCD is observed for complexes with the stoichiometric quantities of porphyrins (4 eq. for TCPPSpm4 and ~12–15 eq. for ZnTCPPSpm4, Appendix A). In sum, the presence of iCD is consistent with strong binding of both porphyrins to the Tel22, and suggests close proximity of the porphyrin ring and G-tetrad(s), indicative of end-stacking. In addition, the split bisignate shape of iCD indicates that porphyrins are not disorderly distributed on Tel22 and that there is communication between the porphyrins in the assembly, in agreement with the RLS data described earlier. The iCD was likewise observed for TCPPSpm4 binding to (dTGGGAG)_4_ GQ aptamer [73] and to poly(dG-dC) and CT DNA [89], and for ZnTCPPSpm4 binding to poly(dG-dC) in both B and Z conformations [90]. However, the shape of the iCD was different from that observed in this work, underlining differences in the binding modes. 

## 4. Materials and Methods 

### 4.1. Porphyrins and Oligonucleotides

TCPPSpm4 and ZnTCPPSpm4 were synthesized as described previously [72,90] and dissolved in double-distilled water (ddH_2_O) at 1–5 mM and stored at 4 °C in the dark. The concentration of TCPPSpm4 was determined via UV-Vis spectroscopy using the extinction coefficient of 3.0 × 10^5^ M^−1^cm^−1^ at 415 nm at pH 6.5 [72]. The extinction coefficient for ZnTCPPSpm4 was measured to be 1.34 × 10^5^ M^−1^cm^−1^ at 424 nm at pH 7 using Beer’s law (Appendix A). Tel22 was purchased from Midland Certified Reagent Company (Midland, TX, USA) and dissolved in 5K buffer (10 mM lithium cacodylate, pH 7.2, 5 mM KCl and 95 mM LiCl). Calf thymus (CT) DNA was purchased from Sigma-Aldrich and dissolved in a solution of 10 mM lithium cacodylate 7.2 and 1 mM Na_2_EDTA at a concentration of 1 mM. The solution was then equilibrated for one week, filtered, and stored at 4 °C. The fluorescently labeled oligonucleotide 5′-6-FAM-GGG(TTAGGG)_3_-Dabcyl-3′ (F21D) was purchased from Integrated DNA Technologies (Coralville, IA, USA), dissolved at 0.1 mM in ddH_2_O, and stored at −80 °C prior to use. The concentrations of all nucleic acids were determined through UV-Vis spectroscopy at 90 °C using the extinction coefficients ε^260 nm^ = 228.5 mM^−1^cm^−1^ for Tel22, 247.6 mM^−1^cm^−1^ for F21D, and 12.2 mM^−1^cm^−1^ (per base pair) for CT DNA. Extinction coefficients were calculated with the Integrated DNA Technologies OligoAnalyzer (available at https://www.idtdna.com/calc/analyzer, accessed on November 20, 2018) which uses the nearest-neighbor approximation model [91,92]. 

To induce GQ structure formation, DNA samples at the desired concentrations alone or in the presence of 1–2 eq. of porphyrin were heated at 95 °C for ten minutes in 5K buffer, allowed to cool to room temperature over three hours, and equilibrated overnight at 4° C. All experiments were done in 5K buffer.

### 4.2. UV-Vis Titrations and Job Plot

UV-Vis experiments were performed on a Cary 300 (Varian) spectrophotometer with a Peltier-thermostated cuvette holder (error of ± 0.3 °C) using 1 cm methylmethacrylate or quartz cuvettes and dual beam detection. The sample cuvette contained 2.3–3.1 µM TCPPSpm4 or 1.0–6.4 µM ZnTCPPSpm4 and the reference cuvette contained 5K buffer. UV-Vis titrations were conducted by adding small volumes of concentrated Tel22 in a stepwise manner to a 1 mL of porphyrin solutions, mixing thoroughly, and equilibrating for at least two minutes. UV-vis scans were collected in the range of 352–500 nm. DNA was added until at least three final spectra were superimposable. All titrations were performed at least three times. All spectra were corrected mathematically for dilutions, and analyzed as described previously using a Direct Fit model [19,42] with GraphPad Prism software at 415 and 429 nm for TCPPSpm4 and 424 nm wavelengths for ZnTCPPSpm4. Job plot UV-Vis titration experiments were performed to independently determine the stoichiometry of ligand-Tel22 binding interactions. Job plot experiments were conducted for both porphyrins using the procedure and data processing described in our earlier work [19]. Both porphyrins and DNA were prepared at 3–4 µM. Job plot experiments were completed at least three times. 

### 4.3. Fluorescence Spectroscopy

#### 4.3.1. Resonance Light Scattering (RLS)

RLS experiments [78] were conducted using a conventional fluorimeter, Fluorolog FL-11 Jobin-Yvon Horiba. A 2.1 mL solution of 2 µM porphyrin in a 1 cm quartz cuvette was titrated with 0.5 mM annealed and equilibrated Tel22 solution at 25 °C. Final concentration of Tel22 varied between 0.05-10.0 µM, and the total volume of all additions was 42 µL (2%). After each addition of Tel22, the cuvette was equilibrated for 10 min and the data was collected with the following parameters: scan range of 380–630 nm, wavelength offset of 0 nm, increment of 1.0 nm, averaging time of 0.5 sec, number of scans 3 (averaged), and 1.5 nm slits for both excitation and emission.

#### 4.3.2. Fluorescent Titrations

Fluorescence titrations were performed on a Photon Technology International QuantaMaster 40 spectrofluorimeter. A 2.0 mL solution of porphyrin in a 1 cm black quartz cuvette was titrated with annealed and equilibrated Tel22 solution at 20 °C. The concentration of TCPPSpm4 was 0.3 µM, and the concentration of ZnTCPPSpm4 was ~0.5 µM. Tel22 was added from three different stocks with increasing concentration: stock 1 was 3–4 µM, stock 2 was 95–150 µM, and stock 3 was 500–850 µM. Total volume of addition was ~60 µL (3%). After each addition of Tel22, the cuvette was equilibrated for at least two minutes and the scan was collected with the following parameters: excitation at 420 nm (at the isosbestic point for TCPPSpm4), emission range of 575–750 nm, increment of 1.0 nm, averaging time of 0.5 sec, one scan, and 3 nm slits both for excitation and emission.

### 4.4. Circular Dichroism (CD) Spectroscopy

CD scans and melting experiments were performed on an Aviv 410 spectropolarimeter equipped with a Peltier heating unit (error of ± 0.3 °C) in 1 cm quartz cuvettes. The solution of 10–15 μM Tel22 was annealed and equilibrated with 2 eq. of porphyrins and CD scans were collected with the following parameters: 220 to 330 nm spectral width, 1 nm bandwith, 1 sec averaging time, 25 °C, and 3–5 scans (averaged). CD melting was performed on the same samples with the following parameters: 294 nm wavelength, 15–90 °C temperature range, 30 sec equilibration time, and 10 sec averaging time. CD scans were collected before and after the melt to check if the melting process is reversible. CD data were analyzed as described in our earlier work [19,42]. 

Two sets of CD titrations were performed. First, 7–15 µM Tel22 was titrated with up to 4 eq. of 0.44 mM TCPPSpm4 or 5.75 mM ZnTCPPSpm4 in 1 eq. increments. After each addition of the porphyrin, the sample was equilibrated for 12 min after which CD scans were collected in 220–330 nm region. Secondly, to detect induced CD signal (iCD) 2–6 µM porphyrin solution was titrated with small increments of 100–200 µM Tel22. Samples were equilibrated for 10 min and CD spectra were collected in the 375–480 nm region using 5–10 scans to obtain good signal-to-noise ratio.

### 4.5. Fluorescence Resonance Energy Transfer (FRET) Assays 

FRET studies were conducted according to the published protocol [81]. A solution of 0.2 µM F21D was incubated in the presence of 0–8 eq. of TCPPSpm4 or 0–20 eq. of ZnTCPPSpm4 and melting curves were collected. FRET competition experiments were performed using 0.2 µM F21D in the presence of fixed amounts of TCPPSpm4 (0.75 µM, 3.7 eq.) or ZnTCPPSpm4 (2.2 µM, 11 eq.) and increasing amounts of CT DNA (up to 96 µM, 480 eq.), and analyzed as described previously [42].

## 5. Conclusions

There is a great need to develop ligands capable of binding to and regulating the stability of GQs strongly and selectively. In this work, we characterized interactions of novel spermine-derivatized porphyrins, TCPPSpm4 and ZnTCPPSpm4, with human telomeric DNA, Tel22. Both porphyrins bind tightly to the GQ with *Ka* of (5–14) × 10^6^ M^−1^ and provide strong stabilization, with the selectivity ratio of 200–300 over dsDNA. Interestingly, we observe a high binding stoichiometry, which may indicate multiple binding modes, the most prominent of which are end-stacking and DNA-assisted self-association of porphyrins. In addition, the spermine arms of the porphyrins likely act as four tentacles reaching into groves and stabilizing the GQ. The mild selectivity for GQ over dsDNA is likely due to strong electrostatic interactions between the polycationic ligand and negatively charged DNA backbone. Consistent with the prior work, addition of Zn(II) to the porphyrin core did not improve selectivity, in spite of the presence of fifth axial water ligand, but increased *Ka* three-fold.

Overall, our findings demonstrate that spermine group derivatization is a valid strategy in the design of novel GQ binders, especially given the fact that polyamines are taken up extensively by cancer cells [67,68], and thus, could be used for selective cancer targeting. Future work will focus on optimizing these porphyrins by decreasing their charge (limiting the number of spermine arms to 1–3) and adding functional groups known to improve GQ selectivity. Biological studies of the new ligands should also be a priority.

## Figures and Tables

**Figure 1 ijms-19-03686-f001:**
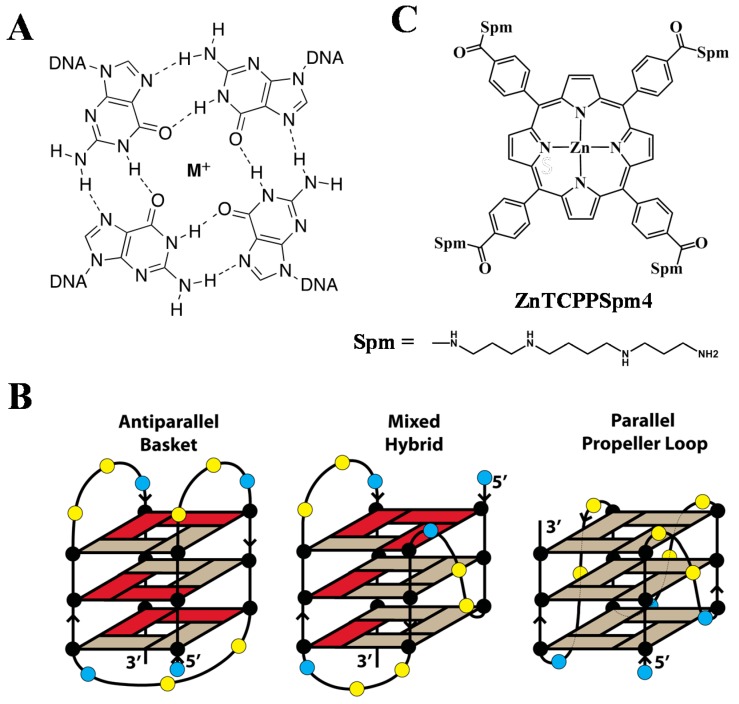
(**A**) Four guanines associate via Hoogsteen hydrogen bonding to form a G-tetrad. M^+^ represents a central coordinating cation, such as Na^+^, K^+^, or NH_4_^+^. (**B**) Schematics of the physiologically-relevant structures of human telomeric DNA, dAGGG(TTAGGG)_3_. Grey and red rectangles represent guanines in *anti* and *syn* conformations. Adenines and thymines are represented as blue and yellow circles, respectively. Strand orientations are depicted with arrows. Mixed-hybrid conformation is that of Form 2. (**C**) Structure of ZnTCPPSpm4; the fifth axial water ligand attached to Zn(II) is not depicted for clarity of the image.

**Figure 2 ijms-19-03686-f002:**
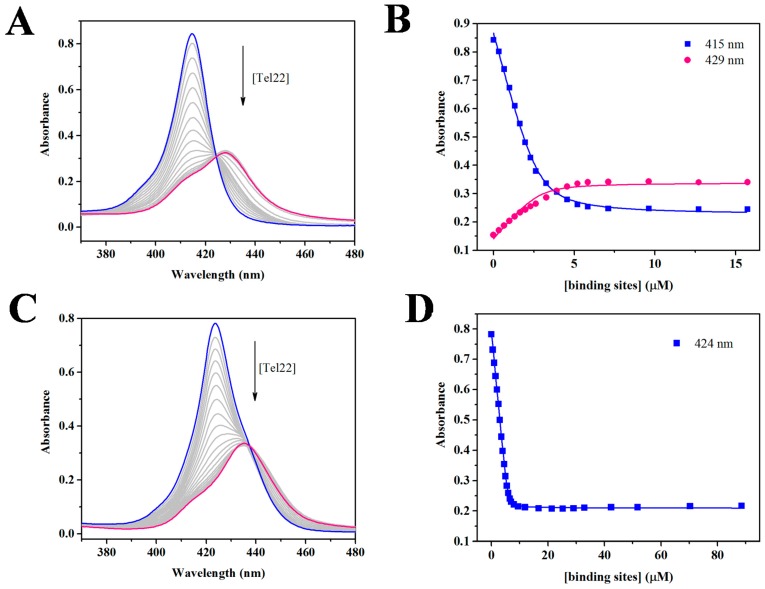
Interactions between porphyrins and Tel22 GQ probed by UV-Vis spectroscopy. (**A**) A representative UV-Vis titration of 2.8 µM TCPPSpm4 with 82.6 µM Tel22. Clear isosbestic point is observed at 424 nm. (**B**) Best fit (solid line) to the titration data monitored at 415 nm (squares) and 429 nm (circles). (**C**) A representative UV-Vis titration of 5.8 µM ZnTCPPSpm4 with 46.3 (followed by 185) µM Tel22. Clear isosbestic point is observed at 442 nm. (**D**) Best fit (solid line) to the titration data monitored at 424 nm (squares). Concentration of binding sites is defined as the concentration of Tel22 multiplied by the binding stoichiometry (4:1 for TCPPSpm4 and 12:1 for ZnTCPPSpm4). Blue lines and points correspond to porphryins alone and pink corresponds to porphyrin-Tel22 complex.

**Figure 3 ijms-19-03686-f003:**
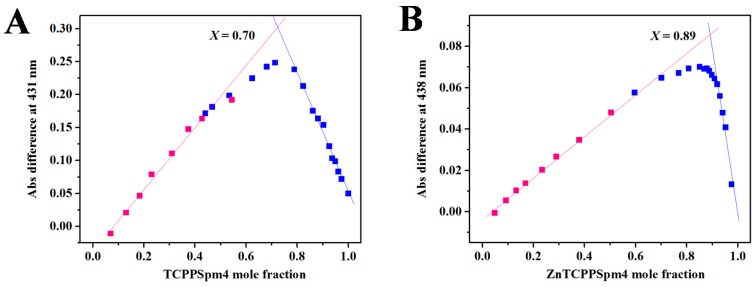
Representative Job plots for (**A**) 3.1 µM TCPPSpm4 and (**B**) 2.9 µM ZnTCPPSpm4 in complex with Tel22 at 25 °C. Porphyrins and Tel22 GQ DNA concentrations were maintained equal within 20%. The Job plots were constructed by plotting the difference in the absorbance values at a specified wavelength vs mole fraction of the porphyrin, *X*. Pink squares represent data collected by titrating porphyrins into DNA; blue squares represent data collected by titrating DNA into porphyrins.

**Figure 4 ijms-19-03686-f004:**
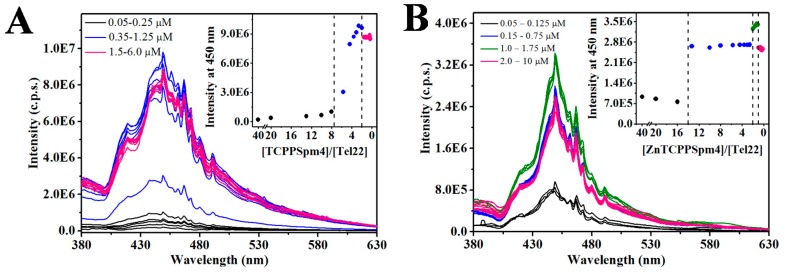
Representative RLS titration of 2.0 µM (**A**) TCPPSpm4 and (**B**) ZnTCPPSpm4 with 500 µM Tel22 at 25 °C. The amounts of Tel22 added are specified in the legend. Inset reports RLS intensity at 450 nm vs [porphyrin]/[Tel22] ratio. Note, the scale in the inset is inverted to follow the progress of the titration which starts with the solution of porphyrin and proceeds toward lower [porphyrin]/[Tel22] ratios.

**Figure 5 ijms-19-03686-f005:**
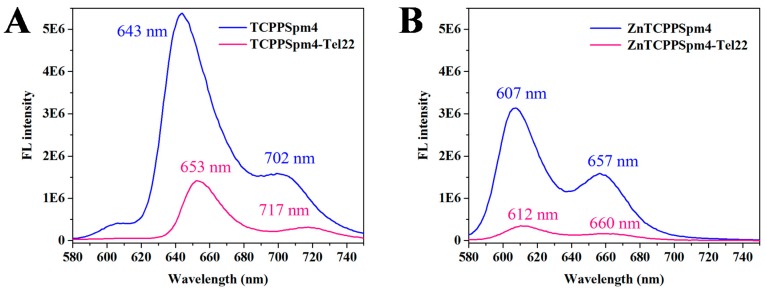
Steady-state fluorescence emission spectra for (**A**) 0.33 µM TCPPSpm4 alone and in the presence of 19.5 fold excess of Tel22 and (**B**) 0.47 µM ZnTCPPSpm4 alone and in the presence of 9.1 fold excess of Tel22. Note, for the ease of comparison, the data were scaled to 1 µM porphyrin.

**Figure 6 ijms-19-03686-f006:**
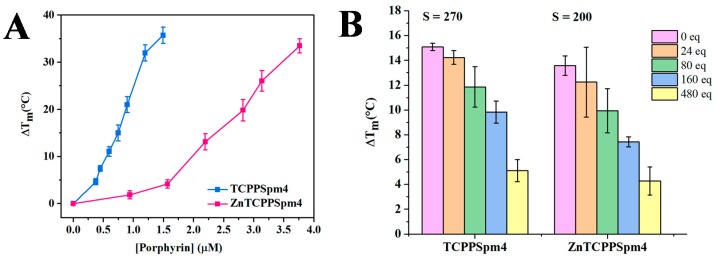
Stabilizing ability and selectivity of TCPPSpm4 and ZnTCPPSpm4 toward human telomeric DNA investigated via FRET. (**A**) Dose dependent stabilization, ΔTm, of 0.2 µM F21D as a function of porphyrin concentration. (**B**) Stabilization of 0.2 µM F21D with 0.75 µM TCPPSpm4 or 2.2 µM ZnTCPPSpm4 in the presence of increasing amount of CT DNA (equivalents relative to F21D are specified in the legend). Concentration of porphyrins was chosen in order to achieve similar starting Tm for the first sample before any CT DNA was added in order to facilitate the comparison. The concentration of F21D is expressed per strand, while the concentration of CT DNA is expressed per base pair. Note, all raw data are presented in Appendix A.

**Figure 7 ijms-19-03686-f007:**
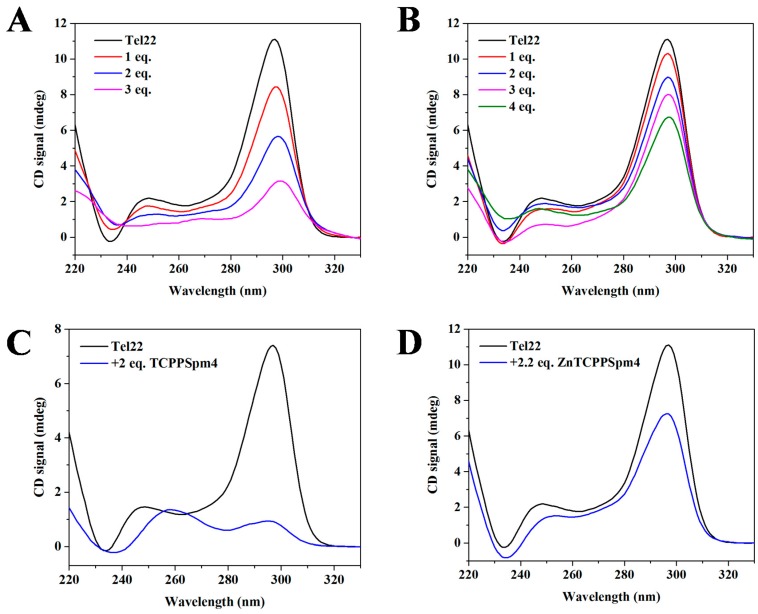
CD titration of 15.0 µM Tel22 with up to 4 eq. of (**A**) TCPPSpm4 and (**B**) ZnTCPPSpm4. Samples were incubated for 12 min after each addition of the porphyrin. CD annealing of (**C**) 10.0 µM Tel22 with 2.0 eq. of TCPPSpm4 and of (**D**) 15 µM Tel22 with 2.2 eq. of Zn TCPPSpm4. Data were collected at 20 °C. We have also completed CD melting on the annealed samples and saw no-to-weak stabilization (Appendix A).

**Figure 8 ijms-19-03686-f008:**
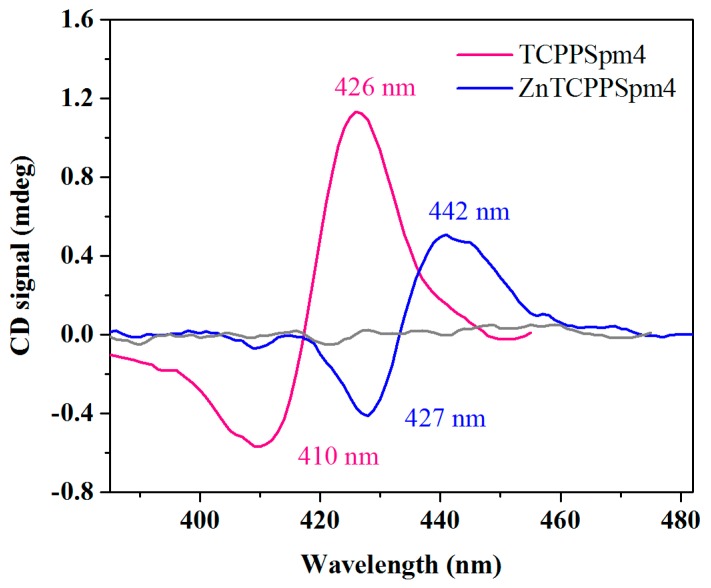
iCD signature of TCPPSpm4-Tel22 and ZnTCPPSpm4-Tel22 complexes prepared at stoichiometric amounts of porphyrins and DNA (4:1 for TCPPSpm4 and 12:1 for ZnTCPPSpm4). The data were scaled to 1 µM porphyrin. The CD scan of porphyrin alone is shown in grey. The data were smoothed using Savitzky–Golay smoothing filter with a 13-point quadratic function.

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
