# Peer review of "Interactions Between Spermine-Derivatized Tentacle Porphyrins and The Human Telomeric DNA G-Quadruplex"

_ijms, 2018, doi:10.3390/ijms19113686_

Round 1
Reviewer 1 Report
We thank the Referees for their positive feedback and useful suggestions. Comments from the referees appear in italic black and our explanations are in blue; new text included in the revised manuscript is in italic blue.
Referee: 1
1. The job plot for the metal-free compound gives a 0.7 break-point, which the authors attribute to a 3:1 stoichiometry (slightly off the 4:1 found by Direct Fit of the UV-vis data). However, this could equally be 2:1 (molar ratio 0.66) in accord with the classic sandwitch/end-stacking mode. Clearly there's a dichotomy between Job plot and Direct Fit, that should be addressed more thoroughly.
One limitation of Job Plot method, as we pointed out in the main text, is its inability to precisely distinguish binding stoichiometries beyond 1:1 and 2:1; as astutely noted by the reviewer, a mole fraction of 0.7 may correspond to either a 3:1 or 2:1 stoichiometry. We have corrected our statement in the main text and provided additional explanation:
“Job plot experiments for TCPPSpm4-Tel22 system yielded an average mole fraction of 0.70 ± 0.04, which corresponds to the binding of 2-3 porphyrins to one Tel22…. In both cases, binding stoichiometries are somewhat lower than those obtained via fitting of the UV-vis titration data. Similar discrepancy was also observed in our previous work where we investigated binding of four different cationic porphyrins to two parallel GQs [1]. While Job plot stoichiometry represents the major binding event, stoichiometry obtained via fitting of UV-vis titration data encompasses strong, weak, and non-specific binding.”
2. In the fluorescence titrations, the authors were unable to determine binding constants, which I assume has to do with the fact that fluorescence from one point on starts going up again. This may be indicative of a change in mechanism above a certain stoichiometry. It would be advised to exclude those points from the data processing, and if the remaining points are insufficient, repeat the experiment with more intermediate points in the early stages of the titration.
We followed the reviewer’s advice but were still unable to obtained acceptable fits to our FL titrations. It is likely due to the fact that the FL data reflects porphyrin binding, porphyrin self-stacking as well as some photobleaching of the signal, although we have modified the parameters to minimize the latter effect. Therefore, we have moved the data from the main text to the Supporting Information (Figure S2). In the main text we have changed the Figure to only contain the FL spectra of porphyrins alone and with saturating amounts of Tel22 and structured the discussion around FL decrease in response to DNA-assisted porphyrin self-stacking.
3. In the CD titration, it is obvious that the destabilization caused by increasing amounts of metal-free porphyrin is such that the G4 structure seems to be entirely disassembled. If this is the case, then the compound would be an unlikely drug candidate to harvest the anticancer potential offered by G-quadruplexes. This is not sufficiently addressed in the text. In relation to this point, the authors have not looked at the DNA peak at 260 nm in their UV-vis titration, which could have provided at least qualitative information on the loss of G4 (and resulting reduction in extinction coefficient).
It is not straightforward to analyse the data from the DNA peak at 260 nm in UV-vis titration due to a) rather significant contribution of porphyrin absorbance in this range, and b) linear increase in DNA concentration modulated by porphyrin binding. The latter effect will make it impossible to determine whether lower Abs at 260 results from DNA unfolding/precipitation or mere binding to the porphyrin. Our lab spent about a year trying to investigate whether Abs changes at 260 nm could serve as a reliable handle on DNA structure and binding to porphyrins (but in this case we titrated porphyrin into DNA) and we concluded that it is not.
We, however, repeated the CD titrations in both kinetic and thermodynamic regime with higher amounts of porphyrins (see Figure 7) to verify possible DNA unfolding. The data indicate that under thermodynamic condition the CD signature of Tel22 is perturbed strongly by TCPPSpm4, but not by ZnTCPPSpm4. Such ligands, however can be useful, because destabilization of biological GQs can be as important as their stabilization under certain conditions. We added the following text to the paper:
“Titration of TCPPSpm4 under kinetic conditions (with short 12 min equilibration) did not alter the conformation of Tel22 but lead to dramatic decrease in the intensity of 295 nm peak at 4-fold excess of the porphyrin (Figure 7A). Under similar conditions, ZnTCPPSpm4 caused only mild decrease in CD signal intensity (Figure 7B). To investigate the system under thermodynamic equilibrium, Tel22 samples were annealed with ~ 2 eq. of porphyrins and equilibrated overnight. The CD signals displayed stronger decrease (Figure 7C,D) in part caused by minor precipitation. Decrease in CD signal intensity was also observed upon interaction of TCPPSpm4 with (dTGGGAG)4 GQ aptamer [2]. Other metallated porphyrins, such as PtTMPyP4 [3], CuTMPyP4, and NiTMPyP4 [4] caused decrease in the intensity of Tel22 CD signal in potassium buffer, while CoTMPyP4 and ZnTMPyP4 did not [4].
Porphyrin-induced decrease in CD signal intensity could be explained, in part, by DNA precipitation, most likely caused by highly charged spermine arms of the porphyrin ligands. The precipitation was minor and was only observed at high porphyrin and DNA concentrations (above 10 µM DNA). In addition, the observed behavior in CD titrations could be explained by preferential binding of porphyrins to single-stranded (ssDNA) which disfavors GQ in the GQ DNA « ssDNA equilibrium. This mode of binding was observed for TMPyP4 [5], triarylpyridines [6], and anthrathiophenedione [7]. However, such data interpretation seems to contradict the observed stabilization of human telomeric DNA in our FRET studies (Figure 6A). Alternatively, we can explain the observed decrease in CD signal intensity by proposing that porphyrins bind to GQ DNA by disrupting and replacing one or more of the G-tetrads, leading unchanged or even enhanced stability. Such explanation reconciles our CD and FRET data and was first proposed by Marchand et al. on a basis of extensive CD and mass-spectroscopy study [8].”
4. In the induced CD, some of the curves don't follow the trend, and due to the very low intensity of the bands and the noise, I consider this data to be rather inconclusive and potentially unnecessary.
We agree with the reviewer that the S/N ratio of the iCD data is low, yet the presence of iCD, even of low intensity, is a clear indication of strong interactions between ligands and Tel22. Moreover, the shape of iCD can be used to determine the binding mode. Split bisignate signal indicates that porphyrins are not disorderly distributed on Tel22 and that there is communication between the porphyrins in the assembly. To confirm this point we also performed Resonance Light Scattering experiments (see new section titled “2.2. Resonance Light Scattering (RLS) Indicates the Formation of Discrete Stoichiometric Porprhyrin-Tel22 Complexes” and new Figure 4 (the Figure and the text are also appended to this document for convenience).
To address the reviewer concern, we moved the iCD titration to SI and left only the iCD spectra at stoichiometric ratios in the main text in order to demonstrate i) the presence of iCD and ii) the shape of iCD signal which we consider important for the story of the paper.
5. All FRET melting curves should be included in the supporting information.
Done, See SI Figure S3
Author Response
uploaded word file

Reviewer 2 Report
1. ... the discussion can be improved by differentiating the effects of the porphyrin and spermine parts of the molecules and providing mechanistic insights.
We agree that differentiating the effects of the spermine and porphyrin core on GQ interactions is of interest. However, all methods used to determine the binding interactions are indirect and differentiation of contributions of spermine arms and porphyrin core is difficult. We added the following clarification in the Conclusion section:
“…we observe a high binding stoichiometry, which may indicate multiple binding modes most prominent of which are end-stacking and DNA-assisted self-association of porphyrins. At the same time, the spermine arms, likely, act as four tentacles reaching into groves and stabilizing the GQ.”
One future experiment, beyond the scope of this manuscript, would be to synthesize porphyrins with 1, 2, and 3 spermine arms and compare binding and spectroscopic data among them to ascertain the precise effect of the spermine groups. Currently we are pursuing the synthesis.
2. Spermine and related polyamines are known to interact with DNA by electrostatic forces, although site-specific interactions have also been reported (Thomas et al. Amino Acids. 2016 Oct;48(10):2423-31; Int J Biol Macromol. 2018 Apr 1;109:36-48; Ouameur and Tajmir-Riahi J Biol Chem. 2004 Oct 1;279(40):42041-54). These interactions provoke structural and conformations alterations, including the induction and stabilization of Z-DNA. There are also many reports on the interaction of porphyrins with DNA. The authors are urged to reference the literature in this area and discuss their findings in the context of previous studies.
We thank the referee for pointing us to the excellent literature sources. We have modified the Introduction extensively with this new information.
“In this work, we focus on two novel tentacle porphyrins, meso-tetrakis-(4-carboxysperminephenyl)porphyrin, TCPPSpm4 and its Zn(II)-derivative, ZnTCPPSpm4, Figure 1C. Binding of tentacle porphyrins to dsDNA is well studied [9-12], but their interactions with GQ DNA remain poorly characterized. We introduced spermine groups to enhance the GQ-binding potential, solubility, and biocompatibility of the porphyrins. Polyamines have been reported to interact with DNA by both electrostatic forces and via site-specific interactions with the phosphate backbone and DNA bases [13-15]. In some cases polyamines induced conformational modifications [16]. Spermine was shown to preferentially bind to the major groove of dsDNA [15]. A variety amines (e.g. pyrrolidine, piperidine, morpholine, 1-ethylpiperazine, N,N-diethylethylenediamine, and guanidine) have been incorporated into GQ ligands, leading to improvements in their GQ binding affinities and water solubility [17-21]. Of equally strong importance, spermine is essential for cellular growth, differentiation [22], and protection against double-strand breaks. Polyamines are currently being exploited as a transport system for cancer drugs due to their well-known ability to accumulate in neoplastic tissues [23-27]. Therefore, we added spermine to meso-tetrakis-(4-carboxyphenyl)porphyrin not only to improve its GQ-binding, but also to facilitate its delivery to cancer cells in future biological studies.”
We also incorporated references to porphyrin-GQ and spermine-DNA binding throughout the text in relevant places.
Author Response
see uploaded word file

Round 2
Reviewer 1 Report
After the corrections done by the authors, I find the quality of presentation to have significantly improved. While most of the methods used to characterize small molecule-G4 interaction are indirect and as such they entail a level of uncertainty, the authors have done very systematic and vigorous work and I feel that now the manuscript is worthy of publication.